# GenKIE: Robust Generative Multimodal Document Key Information Extraction

**Panfeng Cao♠, Ye Wang◇, Qiang Zhang♡, Zaiqiao Meng♣***

♠ University of Michigan
◇National University of Defense Technology
♡Zhejiang University
♣ School of Computing Science, University of Glasgow
`panfengc@umich.edu, wangye19@nudt.edu.cn,`
`qiang.zhang.cs@zju.edu.cn, zaiqiao.meng@glasgow.ac.uk`

## Abstract

Key information extraction (KIE) from scanned documents has gained increasing attention because of its applications in various domains. Although promising results have been achieved by some recent KIE approaches, they are usually built based on discriminative models, which lack the ability to handle optical character recognition (OCR) errors and require laborious token-level labelling. In this paper, we propose a novel generative end-to-end model, named GenKIE, to address the KIE task. GenKIE is a sequence-to-sequence multimodal generative model that utilizes multimodal encoders to embed visual, layout and textual features and a decoder to generate the desired output. Well-designed prompts are leveraged to incorporate the label semantics as the weakly supervised signals and entice the generation of the key information. One notable advantage of the generative model is that it enables automatic correction of OCR errors. Besides, token-level granular annotation is not required. Extensive experiments on multiple public real-world datasets show that GenKIE effectively generalizes over different types of documents and achieves state-of-the-art results. Our experiments also validate the model's robustness against OCR errors, making GenKIE highly applicable in real-world scenarios[1].

## 1 Introduction

The key information extraction (KIE) task aims to extract structured entity information (e.g. key-value pairs) from scanned documents such as receipts (Huang et al., 2019), forms (Jaume et al., 2019), financial reports (Stanisławek et al., 2021), etc. This task is critical to many document understanding applications, such as information retrieval and text mining (Jiang, 2012), where KIE frees the

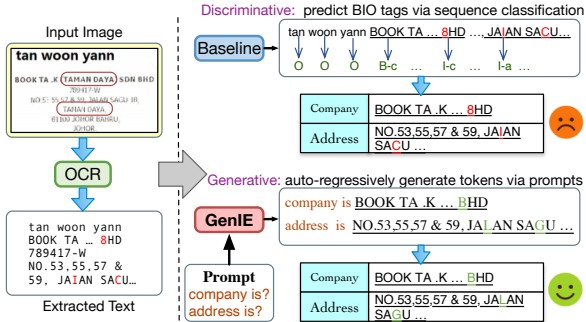

Figure 1: An example of document KIE. Two entity types (i.e. `company` and `address`) are extracted from the document image. The baseline model (Xu et al., 2021) predicts correct labels, but the extracted entity values are still wrong due to the OCR errors (marked in red). GenKIE auto-corrects OCR errors and generates the correct entity values. The multimodal features can be used to distinguish ambiguous entity types (e.g. term 'TAMAN DAYAN' in the red circles of the input image).

business from manually processing a great number of documents and saves a significant amount of time and labour resource (Huang et al., 2019).

The KIE task is often tackled by a pipeline of approaches (Huang et al., 2019), including optical character recognition (OCR) and named entity recognition (NER). The OCR technique, exemplified by Tesseract[2], is employed to discern textual and layout attributes from the input images, namely the scanned documents. The NER model (Mohit, 2014) is used to discriminatively extract salient details from the derived texts, such as pinpointing specific entities from the text and layout features based on the annotated beginning-inside-outside (BIO) tags via a sequence classification strategy. A plethora of methodologies adhering to this framework have emerged, leveraging a synergy of multimodal features—textual, visual, and layout data—from the document image. Notably, contemporary KIE models like StrucText (Li et al., 2021), BROS (Hong et al., 2020), and LayoutLMv2 (Xu et al., 2021) exhibit commendable results through

---

*Corresponding author.

[1]Our code and pretrained model are publicly available at `https://github.com/Glasgow-AI4BioMed/GenKIE`.

[2]`https://tesseract-ocr.github.io/`

the use of multimodal pretrained frameworks.

However, one limitation of these models is that they highly rely on the OCR model to extract texts from scanned documents and inevitably suffer from OCR errors. These OCR errors in texts will eventually render wrong entity information. As shown in Figure 1, although the classification-based model tags the entities correctly, the result is still wrong due to the OCR errors. Another limitation is that the semantic ambiguity in the document is hard to be captured by the existing models. For example, as shown in Figure 1, two image patches of the same text (TAMAN DAYAN) in the receipt have different entity types. It is difficult for some existing approaches to improve performance by using only textual information. Other signals such as layout and visual information play a critical role in identifying the correct entity type (Xu et al., 2021). Therefore effective incorporation of multimodal features is indispensable to improve the model performance for the KIE task.

To cope with the mentioned problems, we propose GenKIE, a robust multimodal generative model for document KIE. GenKIE utilizes the encoder-decoder Transformer (Vaswani et al., 2017) as the backbone and formulates document KIE as a sequence-to-sequence generation task. The encoder effectively incorporates multimodal features to handle semantic ambiguity and the decoder generates the desired output autoregressively following the carefully designed template (i.e. prompt) and auto-correcting OCR errors. An example is shown in Figure 1 to demonstrate the prompting technique, the label *company* in the prompt guides the model to generate words that correspond to company names. Other labels such as *address* serve as additional label semantic signals and provide shared context about the task. Thanks to the generation capability, GenKIE does not need the laborious granular token-level labelling that is usually required by discriminative models.

Extensive experiments demonstrate that our proposed GenKIE model has not only achieved performance levels comparable to state-of-the-art (SOTA) models across three public KIE datasets but also exhibits enhanced robustness against OCR errors. Our contributions are summarized as follows:

- We propose GenKIE, a novel multimodal generative model for the KIE task that can generate entity information from the scanned documents auto-regressively based on prompts.

- We propose effective prompts that can adapt to different datasets and multimodal feature embedding that deals with semantic ambiguity in documents. Our model generalizes on unseen documents with complex layouts.

- Extensive experiments on real-world KIE datasets show that GenKIE has strong robustness against OCR errors, which makes it more applicable for practical scenarios compared to classification-based models.

## 2 Related Works

### 2.1 Conventional KIE Methods

Traditional document KIE methods (Dengel and Klein, 2002; Schuster et al., 2013) depend on predefined templates or rules to extract key information from scanned documents. Due to the extensive manual effort and specialized knowledge required to design these specific templates for each entity type, these approaches are not suitable for effectively managing unstructured and intricate document layouts. Later KIE systems formalize the problem as an NER task and start to employ powerful machine learning models. For example, the BiLSTM-CRF model employed by Huang et al. (2015); Lample et al. (2016); Ma and Hovy (2016); Chiu and Nichols (2016) decodes the chain of entity BIO tags from either textual or textual and visual features. Katti et al. (2018) proposes an image-based convolutional encoder-decoder framework that can encode the semantic contents. The graph-based LSTM utilized by Peng et al. (2017); Song et al. (2018) allows a varied number of incoming edges in a memory cell to learn cross-sentence entity and relation extraction. While those models are effective in their domains, they do not make use of all multimodal features available in the documents and thus could not solve semantic ambiguity and generalize. Therefore, recent research emphasizes more on the incorporation of multimodal features to generalize on documents with varied and complicated layouts. For example, PICK (Yu et al., 2021) models document input as a graph, where the text and visual segments are encoded as nodes and spatial relations are encoded as edges. However, due to the lack of pretraining on a large corpus, those methods are relatively limited in terms of robustness and generalization ability.

## 2.2 Multimodal-based KIE Methods

Multimodality-based transformer encoder models, which are pretrained on large-scale linguistic datasets, show strong feature representation and achieve SOTA performance in downstream KIE tasks. LayoutLM (Xu et al., 2020) first proposes the pretraining framework to handle document information extraction. Textual and layout features are jointly utilized in pretraining and visual features are embedded in finetuning. LayoutLMv2 (Xu et al., 2021) further improves LayoutLM by incorporating visual features in pretraining. LayoutLMv3 (Huang et al., 2022) introduces a new word-patch alignment objective in pretraining, which reconstructs the masked image tokens from surrounding text and image tokens. DocFormer (Appalaraju et al., 2021) designs a multimodal self-attention layer to facilitate multimodal feature interaction. BROS (Hong et al., 2020) is also an encoder-based model that utilizes graph-based SPADE (Hwang et al., 2021) classifier to predict both entity tags and entity relations. Layout features are specifically exploited to solve the reading order serialization issue. Similar to BROS, LAMBERT (Garncarek et al., 2021) augments the input textual features with layout features to train a layout-aware language model. StrucText (Li et al., 2021) and StrucTexTtv2 (Yu et al., 2023) exploit the structured information from the document image and use it to aid entity information extraction. However, all the above-mentioned models are classification-based, which means fine-grained annotations are necessary and they lack the mechanism to auto-correct OCR errors.

## 2.3 Prompt-based Language Models

Recently the paradigm of "pretrain, prompt, and predict" has been prevailing in NLP due to its high adaptability and effectiveness in downstream tasks (Liu et al., 2023). Our proposed work follows this new paradigm by using OFA (Wang et al., 2022a), a multimodal generative vision language model, as the model backbone and leveraging task-specific prompts in finetuning. It is worth mentioning TILT (Powalski et al., 2021), which is also a generative KIE model. The major difference is that GenKIE emphasizes the prompt design and the model's OCR correction capability in practical scenarios, which are not explored in TILT. Donut (Kim et al., 2022) is an OCR-free model and by design unaffected by OCR errors. It encodes the image features with a Swin Transformer (Liu et al., 2021) and decodes the key information with a Bart (Lewis et al., 2020)-like decoder. However, the limitation of Swin Transformer to capture the local character patterns might lead to sub-optimal KIE results.

## 3 Task Formulation

In this work, we address the KIE task which supports many downstream applications such as entity labeling (Jaume et al., 2019) and entity extraction (Huang et al., 2019). In particular, given a collection of scanned document images $\mathcal{I_D}$, the goal of KIE is to extract a set of key entity fields (i.e. key-value pairs) $\mathcal{K} = \{< T_i, V_i > | 1 \leq i \leq N_I\}$ for each image $I \in \mathcal{I_D}$, where $T_i$ is the predefined entity type, $V_i$ is the entity value, and $N_I$ is the number of entity fields in image $I$. The entity types of each image example can be given or not, depending on the downstream applications. For example, the output of the entity extraction task is the entity values of predefined entity types, while the output of the entity labelling task is the extracted entities and their entity types.

## 4 Methodology

The overall architecture of GenKIE is shown in Figure 2. Unlike the encoder-based models (e.g. LayoutLMv2 (Xu et al., 2021)), GenKIE is a generative model that uses a multimodal encoder-decoder model, i.e. OFA (Wang et al., 2022a), as the backbone. The *encoder* of GenKIE embeds multimodal features (e.g. text, layout and images) from the input and the *decoder* generates the textual output by following the prompts. Entity information is parsed from the decoder output. In the following sections, we illustrate the process of multimodal feature embedding, and then go over the techniques of prompting and inferring.

### 4.1 Encoder

Following the common practice of the KIE task (Xu et al., 2020, 2021), we use an off-the-shelf OCR tool to extract the textual and layout features (i.e. transcripts and bounding boxes of the segments) from the input image. Then we use our backbone language encoder (i.e. byte-pair encoding (BPE) (Sennrich et al., 2016)), layout encoder and visual encoder (i.e. ResNet (He et al., 2016)) to obtain embeddings from these features.

#### 4.1.1 Textual Embedding

To process the textual feature, we apply the BPE tokenizer to tokenize the text segment into a sub-

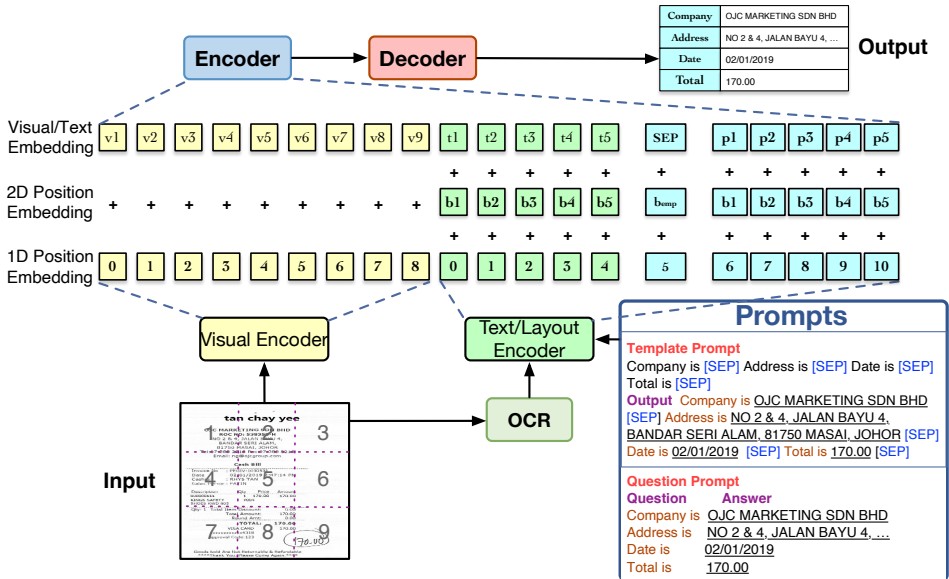

Figure 2: An illustration of GenKIE performing entity extraction from a scanned SROIE receipt with the template and QA prompts. The input to the encoder consists of patched visual tokens and textual tokens embedded along with the positional features. Following the prompts, the decoder generates the desired output, which is processed into four entity key-value pairs.

word token sequence and then wrap around the sequence with the start indicator tag [BEG] and the end indicator tag [END]. Then a sequence of prompt tokens is appended after the [SEP] tag, which marks the end of the transcript tokens. Extra [PAD] tokens are appended to the end to unify the sequence length inside the batch. The token sequence $S$ is formulated as:

$$S = [\text{BEG}], \text{BPE}(T), [\text{SEP}], \text{BPE}(P), [\text{END}], ..., [\text{PAD}], \quad (1)$$

where $T$ represents the transcripts and $P$ represents the prompt. To preserve positional information, we combine the token embedding with the trainable 1D positional embedding in an element-wise manner to obtain the final textual embedding. Specifically, the $i$-th textual token embedding $TE_i$ is represented as:

$$TE_i = \text{EMB}(S_i) + \text{POSEMB1D}(i) \in \mathbb{R}^d, i \in [0, L], \quad (2)$$

where $d$ is the embedding dimension, $L$ is the sequence length, $\text{EMB} : \mathbb{R} \to \mathbb{R}^d$ is the token embedding layer shared between encoder and decoder. $\text{POSEMB1D} : \mathbb{R} \to \mathbb{R}^d$ is the 1D positional embedding layer that is not shared.

### 4.1.2 Layout Embedding

GenKIE uses a layout embedding layer to capture the spatial context of text segments. We first normalize and discretize all bounding box coordinates so they fall between [0, 1024). In this

work, we use a tuple of normalized coordinates to represent the layout feature of the tokens in that bounding box. For instance, the layout feature of $i$-th bounding box can be represented by $b_i = (x_0^i, x_1^i, w_i, y_0^i, y_1^i, h_i)$, where $(x_0^i, y_0^i)$ is the left top coordinate, $(x_1^i, y_1^i)$ is the bottom right coordinate, $w_i$ is the width and $h_i$ is the height of the bounding box, and all textual tokens in the bounding box share the same layout feature. Special tokens such as [BEG], [END] and [SEP] default to have empty feature $b_{emp} = (0, 0, 0, 0, 0, 0)$. If the prompt token $p_j$ refers to the same textual token $t_i$, then $b_i$ is shared with $p_j$ for prompt layout embedding, otherwise $p_j$ defaults to use $b_{emp}$. The final encoder layout embedding is given by:

$$LE_i = \text{CONCAT}(\text{POSEMB2D}_x(x_0^i, x_1^i), \text{POSEMB2D}_w(w_i),$$
$$\text{POSEMB2D}_y(y_0^i, y_1^i), \text{POSEMB2D}_h(h_i)) \in \mathbb{R}^d, \quad (3)$$

where CONCAT is the concatenation function; $\text{POSEMB2D}_x$, $\text{POSEMB2D}_y$, $\text{POSEMB2D}_w$ and $\text{POSEMB2D}_h$ are linear transformation to embed spatial features correspondingly following (Xu et al., 2021). The two-axis features are concatenated to form the 2D spatial layout embedding.

### 4.1.3 Visual Embedding

For visual embedding, we first resize the input image $I$ to $480 \times 480$ and then use a visual encoder consisting of the first three blocks of ResNet (He et al., 2016) following the common practice of the

| Dataset | Prompt Type | Prompt | Generation Target | Example Target |
|---------|-------------|--------|-------------------|----------------|
| SROIE | Template (All Types) | **type1** is [SEP] 
 **type2** is [SEP] 
 ... | **type1** is **value1** [SEP] 
 **type2** is **value2** [SEP] 
 ... | **company** is **yongfatt enterprise** [SEP] 
 **address** is **no 122 jalan dedap johor bahru** [SEP] 
 ... |
| SROIE | Template | **type** is [SEP] | **type** is **value** [SEP] | **company** is **yongfatt enterprise** [SEP] |
| SROIE | Question | **type** is? | **value** | **yongfatt enterprise** |
| FUNSD | Template | **value** is [SEP] | **value** is **type** [SEP] | **coupon code registration form** is **header** [SEP] |
| FUNSD | Question | **value** is? | **type** | **header** |
| CORD | Template | **value** is [SEP] | **value** is **type** [SEP] | **Es Kopi Rupa** is **menu.nm** [SEP] |
| CORD | Question | **value** is? | **type** | **menu.nm** |

Table 1: Prompting techniques for different datasets. For the template prompt, GenKIE fills in the missing entity information in the template and each entity type and value pair is ended with the [SEP] indicator. Specifically for the SROIE dataset, the template prompt can include all entity types, for which GenKIE generates all entity values in one go. For the question prompt, GenKIE generates the answer, which is similar to the question-answering task.

vision language models (Wang et al., 2022b,a) to extract the fix-sized contextualized feature map $M \in \mathbb{R}^{H \times W \times C}$, where $H$ is the height, $W$ is the width and $C$ is the number of channels. The feature map $M$ is further flattened into a sequence of patches $Q \in \mathbb{R}^{K \times d_{img}}$, where $d_{img}$ is the image token dimension and $K = \frac{H \times W}{P^2}$ is the image token sequence length given a fixed patch size $P$. The patches are fed into a linear projection layer to conform to the textual embedding dimension. Similarly to textual embedding, we also add trainable 1D positional embedding to retain the positional information since the visual encoder does not capture that. The visual embedding is formulated as:

$$V E_i = Q_i \cdot \mathbf{W} + \text{PosEmb1D}(i) \in \mathbb{R}^d, \quad (4)$$

where $\mathbf{W}$ is the trainable parameters of a linear projection layer that maps from the image token dimension to the model dimension. We concatenate the image embedding with textual embedding to produce the final document multimodal feature embedding:

$$E = \text{Concat}(V E, T E + L E) \in \mathbb{R}^{(T+L) \times d}, \quad (5)$$

where the concatenation is performed in the second to the last dimension.

### 4.2 Prompts

Prompts are a sequence of text inserted at the end of the encoder input to formulate the task as a generation problem. The decoder is provided with the filled-in prompt as the sequence generation objective. Inspired by the unimodal textual prompts in DEGREE (Hsu et al., 2022), we design simple and efficient spatial awareness prompts which utilize both textual features and layout features. The

prompts go through the same textual and layout embedding steps as the transcripts (see Figure 2).

We introduce two types of prompts, template prompt and question prompt as presented in Table 1. The performance of different prompts is discussed in the ablation study (see §6.3.2). For the entity extraction task, the prompt is designed to be prefixed with the desired entity type and GenKIE continues the prompt with the entity value. For example, in Figure 2, GenKIE generates the value of the Company type by answering the question prompt "Company is?". For the entity labeling task, the prompt is prefixed with the entity value and GenKIE generates the entity type similarly to the classification-based model. For example, in Table 1 on the CORD dataset, GenKIE generates the entity type of Es Kopi Rupa by filling in the template "Es Kopi Rupa is [SEP]".

In essence, the prompt defines the decoder output schema and serves as the additional label or value semantic signal to enforce the model to generate the expected output. Although prompt engineering requires manual efforts to construct appropriate prompts, it is more effortless than granular token-level labelling for the entity extraction task. Unlike vectorized prompts in previous works (Li and Liang, 2021; Yang et al., 2022), we design the prompts in natural sentences to leverage the power of the pretrained decoder. Moreover, the usage of natural sentences further reduces the overhead of composing the prompt. See Appendix C for the method to construct the prompt.

### 4.3 Decoder

The decoder input has the filled-in prompt that serves as the learning target. The same BPE tokenizer and textual embedding in the encoder are

utilized. Note that only the textual modality is leveraged to take advantage of the pretrained decoder.

During inference, if the prompt is a template, we utilize prefix beam search that constrains the search space to start with the template prefix. For example, in Table 1, for the template prompt of the FUNSD dataset, we directly search after the prefix "`coupon code registration form` is" for the entity type without generating the prefix from scratch. Prefix beam search not only makes the inference faster and more efficient but also guarantees a deterministic output that follows the template. Entity information can then be parsed from the template easily. In our experiments, we observe prefix beam search can improve the model performance on all datasets. We will discuss more in the ablation study.

## 5 Experimental Setup

### 5.1 Settings

GenKIE takes a multimodal encoder-decoder-based pretrained model as its backbone. In particular, the backbone model weights used in this work are initialized from the pretrained OFA base model (Wang et al., 2022a), which consists of a 6-layer Transformer encoder and a 6-layer Transformer decoder. The model dimension is 768. For the entity extraction task on the SROIE dataset, the maximum sequence length is 512. For the entity labelling task on the CORD and FUNSD datasets, the maximum sequence length is 32 for the question prompt and 128 for the template prompt. The maximum encoder sequence length is set to 1024 across all datasets. The model is trained for 50 epochs with a batch size of 64 and an initial learning rate of $5e - 5$. For beam search during inference, the number of beams is configured as 5 and we limit the maximum generated sequence length to 512 for SROIE and 128 for CORD and FUNSD.

### 5.2 Datasets and Baselines

We conduct experiments on three real-world datasets, including SROIE (Huang et al., 2019), CORD (Park et al., 2019) and FUNSD (Jaume et al., 2019). Table 2 shows statistics of these datasets[3].

Eleven baselines are used for comparison, including the SOTA models such as LayoutLMv3 (Huang et al., 2022), LayoutLMv2 (Xu et al., 2021), DocFormer (Appalaraju et al., 2021) and TILT (Powal-

---

[3]See Appendix A for more details about these datasets.

| Dataset | Type | # Keys | # Images |
|---------|------|--------|----------|
| FUNSD | Form | 4 | Train 149, Val 0, Test 50 |
| SROIE | Receipt | 4 | Train 626, Val 0, Test 347 |
| CORD | Receipt | 30 | Train 800, Val 100, Test 100 |

Table 2: Statistics of our used datasets.

ski et al., 2021). All baselines except TILT are classification-based models.

## 6 Results and Analysis

This section provides experimental results and analyses of our model's effectiveness (§6.1) and robustness against OCR errors (§6.2) on the KIE task. An ablation study is conducted to analyze the contributions of each component of our model (§6.3). Finally, we provide a case study in §6.4.

### 6.1 KIE Effectiveness

We evaluate the effectiveness of GenKIE on the entity extraction task (using the SROIE dataset) and the entity labelling task (using the FUNSD and CORD datasets). In this paper, we use token-level evaluation metrics across all datasets.

As shown in Table 3, GenKIE outperformed LayoutLMv3 by a certain margin on the FUNSD dataset. However, GenKIE demonstrates comparable performance to other strong encoder-based baselines, such as LayoutLMv2 (Xu et al., 2021), DocFormer (Appalaraju et al., 2021), and LAMBERT (Garncarek et al., 2021). This result validates the effectiveness of our method and suggests that modelling the KIE task in a generative manner is nearly as capable as classification-based models.

From Table 3, we can also observe that models with more modality features generally perform better than those with fewer across all datasets. This suggests that integrating multimodal features can effectively improve KIE performance.

### 6.2 Model Robustness

To verify the robustness of GenKIE against OCR errors, we run entity extraction experiments on the SROIE dataset with simulated OCR errors. We choose LayoutLMv2 (Xu et al., 2021) as our baseline model because of its similar embedding design.

We manually add OCR errors to the original receipt transcripts by replacing the words with visually similar ones[4], e.g. `hello` v.s. `he11o`. Each

---

[4]The visually similar characters are collected from errors produced by the OCR tool.

| Model | Modality | # Params | SROIE | | | CORD | | | FUNSD | | |
|---|---|---|---|---|---|---|---|---|---|---|---|
| | | | P | R | F | P | R | F | P | R | F |
| BERT (Devlin et al., 2019) | T | 110M | 90.99 | 90.99 | 90.99 | 88.33 | 91.07 | 89.68 | 54.69 | 67.10 | 60.26 |
| RoBERTa (Liu et al., 2019) | T | 125M | 91.07 | 91.07 | 91.07 | - | - | - | 66.48 | 66.48 | 66.48 |
| UniLMv2 (Bao et al., 2020) | T | 110M | 94.59 | 94.59 | 94.59 | 89.87 | 91.98 | 90.92 | 65.61 | 72.54 | 68.90 |
| BROS (Hong et al., 2020) | T+L | 110M | 94.93 | 96.03 | 95.48 | 95.58 | 95.14 | 95.36 | 81.16 | 85.02 | 83.05 |
| LayoutLM (Xu et al., 2020) | T+L | 113M | 94.38 | 94.38 | 94.38 | 94.37 | 95.08 | 94.72 | 76.77 | 81.95 | 79.27 |
| LAMBERT (Garncarek et al., 2021) | T+L | 125M | - | - | 96.93 | - | - | 94.41 | - | - | - |
| LayoutLMv2 (Xu et al., 2021) | T+L+V | 200M | 96.25 | 96.25 | 96.25 | 94.53 | 95.39 | 94.95 | 80.29 | 85.39 | 82.76 |
| LayoutLMv3 (Huang et al., 2022) | T+L+V | 133M | 94.91 | 95.68 | 95.30 | - | - | **96.56** | - | - | **90.29** |
| StrucText (Li et al., 2021) | T+L+V | 107M | 95.84 | **98.52** | 96.88 | - | - | - | **85.68** | 80.97 | 83.09 |
| TILT (Powalski et al., 2021) | T+L+V | 230M | - | - | **97.65** | - | - | 95.11 | - | - | - |
| DocFormer (Appalaraju et al., 2021) | T+L+V | 183M | - | - | - | **96.52** | **96.14** | 96.33 | 80.76 | **86.09** | 83.34 |
| GenKIE | T+L+V | 180M | **97.40** | 97.40 | 97.40 | 95.75 | 95.75 | 95.75 | 83.45 | 83.45 | 83.45 |

Table 3: Overall performance of the compared models on our three datasets. Bold indicates the best performance per metric and underline indicates the second best. In the modality column, **T** represents the textual modality, **L** represents the layout modality and **V** represents the visual modality. All models except GenKIE and TILT are discriminative. For LayoutLMv3, the evaluation result on the SROIE dataset is obtained by using the source code provided by the authors. Other performance metrics of compared models are obtained from their original papers.

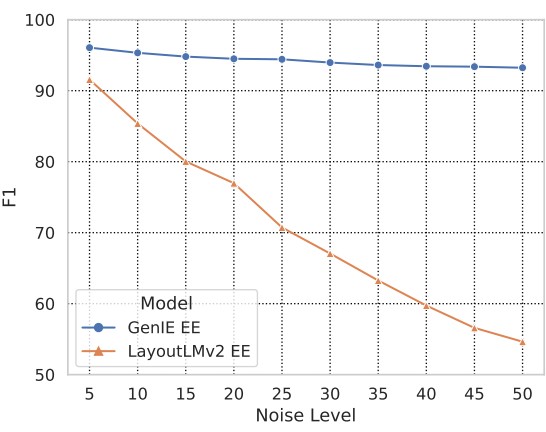

Figure 3: Model robustness experiments under different noise levels. LayoutLMv2 entity extraction is heavily impacted by the OCR errors. Under the 50% error level, the F1 score drops around 40%. While the performance of LayoutLMv2 entity labelling and GenKIE entity extraction drops only around 3%.

word in the transcript has $n\%$ of the chance to be replaced ($n$ denotes the level of OCR errors) and if there are no visually similar characters, we keep the original text.

We run experiments under different levels of OCR errors ranging from 5% to 50% with a 5% step. As shown in Figure 3, when the error level increases, the F1 score of LayoutLMv2 drops significantly since the OCR errors are not handled by design. Under the 50% OCR error level, GenKIE still achieves nearly 94% F1 score dropping only 3% from 97%, which proves that GenKIE has strong robustness against OCR errors. However, it is worth

| Modality | Precision | Recall | F1 |
|---|---|---|---|
| T | 96.24 | 96.24 | 96.24 |
| T + V | 97.20 | 96.82 | 97.01 |
| T + L | 96.89 | 97.39 | 97.14 |
| T + L + V | **97.40** | **97.40** | **97.40** |

Table 4: Model performance of different modalities for entity extraction on the SROIE dataset. Bold indicates the best performance.

noting that GenKIE can generate wrong answers although there are no OCR errors in the input. We present some qualitative examples of KIE on the SROIE dataset in 6.4 and Appendix B.

### 6.3 Ablation Study

#### 6.3.1 Effectiveness of Multimodality Features

For the entity extraction task on the SROIE dataset, we ran experiments with the question prompt to analyze the feature importance of the layout and visual embedding. The textual modality in our model is vital and cannot be removed.

As is shown in Table 4, the model trained with full modality achieves the highest F1 score. The model trained with unimodal textual modality has the lowest score, which verifies that both visual and layout embeddings are effective in this task.

#### 6.3.2 Effectiveness of Different Prompts

In Table 5, we compare the effectiveness of different prompts on all datasets. In particular, for entity extraction on the SROIE dataset, we have an additional template prompt that includes all entity types in the prompt, e.g. the template prompt in Figure

| Dataset | Prompt | Prefix Search | P | R | F |
|---|---|---|---|---|---|
| SROIE | Template (All Types) | - | 96.88 | 96.96 | 96.92 |
| | Template | ✓ | 96.76 | 96.99 | 96.87 |
| | Template | ✗ | 96.86 | 96.63 | 96.75 |
| | QA | - | 97.40 | 97.40 | **97.40** |
| FUNSD | Template | ✓ | 83.45 | 83.45 | **83.45** |
| | Template | ✗ | 76.42 | 76.42 | 76.42 |
| | QA | - | 75.17 | 75.17 | 75.17 |
| CORD | Template | ✓ | 95.75 | 95.75 | **95.75** |
| | Template | ✗ | 94.24 | 94.24 | 94.24 |
| | QA | - | 92.44 | 92.44 | 92.44 |

Table 5: Effectiveness of different prompts and prefix beam search on SROIE, FUNSD, and CORD datasets. Bold indicates the best F1 performance over different schemes per each dataset.

2. The model needs to fill in all the required entity values in one generation. Note that formulating the dataset with the prompt of a single entity type can result in possible duplication of data as the document can have multiple entity types. It requires careful processing to not incur unnecessary computational overhead during training. However, we do not observe a large performance gap between a template prompt with all entity types and other prompts, e.g. 96.92 v.s. 97.40 in terms of F1, which suggests that using a template prompt with all entity types is a simple while efficient mechanism to avoid duplication data processing at the cost of a minor performance drop.

For entity labeling on the CORD and FUNSD datasets, the template prompt outperforms the question prompt while for entity extraction on the SROIE dataset, the question prompt outperforms the template prompt. This indicates in the entity extraction task, the model benefits more from the question-answering formulation since the answer space is unconstrained from the template and the generation capability is utilized more. In the entity labelling task, GenKIE essentially works similarly to classification-based models. The value semantics provided by the template prompt can effectively restrict the search space and guide the model to generate the desired entity type.

### 6.3.3 Effectiveness of Prefix Beam Search

As presented in Table 5, when the template prompt is used, prefix beam search outperforms vanilla beam search by a small margin on the SROIE and CORD datasets, while the performance gap is notably large on the FUNSD dataset (e.g. 83.45 v.s.

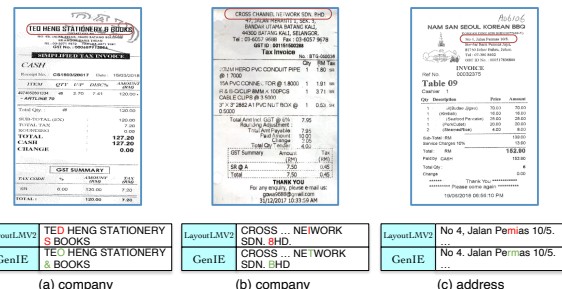

Figure 4: Qualitative examples of the entity extraction task on SROIE. While LayoutLMv2 was misled by the OCR result, GenKIE corrects these OCR mistakes.

76.42). This could be due to more abundant value semantics in the entity labeling task of the FUNSD datasets, which adds difficulty for the model to generate the complete prompt from scratch in vanilla beam search. In the constrained search space of prefix beam search, the model only needs to generate the entity type.

### 6.4 Case Study

The motivation behind GenKIE is to cope with the OCR errors for practical document KIE. To verify the model's robustness and effectiveness, we show some qualitative examples of the output of GenKIE on the SROIE dataset. As presented in Figure 4, subfigures (a) to (c) have multiple OCR errors in the company and address entities, and our GenKIE is able to correct all of them, e.g. `TED` v.s. `TEO`, `8HD` v.s. `BHD`, `Pemias` v.s. `Permas`, etc. See Appendix B for more examples, including the failure cases where GenKIE generates the wrong entities even when there is no OCR error.

## 7 Conclusion

We propose GenKIE, a novel prompt-based generative model to address the KIE task. GenKIE is effective in generating the key information from the scanned document images by following carefully designed prompts. The validated strong robustness against the OCR errors makes the model applicable in real-world scenarios. Extensive experiments over two KIE tasks (i.e. entity labelling and entity extraction) on three public cross-domain datasets demonstrate the model's competitive performance compared with SOTA baselines. Our GenKIE incorporates multimodal features, which enables the integration with other vision language models, offering possibilities for future exploration and experimentation.

## Limitations

One limitation with GenKIE is that it might require additional processing of the datasets. As most of the document KIE datasets nowadays might be tailored for the classification task only, it takes some time to formulate the datasets with different prompts. And it also takes time to experiment with those prompts to find the best one.

Besides, although the multimodal feature embedding is effective in coping with semantic ambiguity, it is only utilized in the encoder in finetuning. Pretraining the model on large document datasets with multimodal features could potentially improve the model's performance.

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

## A  Datasets

**FUNSD** consists of 199 forms annotated with 4 entity types, where each entity type can correspond to multiple values. Each image of the dataset contains many value-type pairs and the task to perform is entity labeling.

**SROIE** contains 626 receipts for training and 347 receipts for testing and each receipt has four entity types for information extraction. Unlike FUNSD, SROIE focuses on key entity extraction from the document, where there could be a lot of unrelated entities. The model performs entity extraction on this dataset and is evaluated on the mentioned four entity types.

**CORD** has 800 scanned receipts for the training set, 100 for the validation set, and 100 for the test set. There are in total 4 main categories in this dataset, which are further classified into 30 subcategories, such as menu name under menu category, total price under total category etc. Similar to the FUNSD dataset, the task is entity labelling.

## B  Case Study

We provide more qualitative examples on the SROIE dataset to validate the effectiveness of GenKIE. In subfigure (d), an entire word **IPOH**

is intentionally removed from the end of the address line, leaving OCR to recognize a blank string, GenKIE is still able to reconstruct the word thanks to the powerful generative capability. It's worth mentioning that subfigures (e) and (f) are failure cases, in which GenKIE generates wrong entity values even if the OCR result is correct. This is reasonable in that OCR errors could influence model training and lead the model to learn wrong features.

## C  Prompt Construction

The prompt is intended to be precise and simple to not incur any semantic ambiguity and linguistic overhead. One strategy to construct the prompt for the entity extraction task is first identifying all the entity types in the document. Then for each entity type, e.g. A, we can either construct the template prompt such as "A is [SEP]" and "A: [SEP]" or the question prompt such as "What is A?" and "A is ?". And we append the prompt to the end of the document transcript to form a training instance. For the entity labeling task, we use the entity value to construct the prompt and the same strategy can be applied.

## D  Zero-Shot and Few-Shot Entity Extraction

To further test our model's generalization ability, we conduct entity extraction experiments under zero-shot and few-shot settings on the SROIE dataset.

### D.1  Experimental Settings

In each experiment, we select one entity type as an unseen type and the other types as common types. To simulate the zero-shot setting, we remove all training instances with unseen types. For the few-shot setting, we only keep $k$ training instances for the unseen type (denoted as $k$-shot). We evaluate the performance only for those unseen types in the test dataset with the F1 score.

### D.2  Experimental Results

Table 6 shows the results of zero/few-shot experiments. Performance of GenKIE is relatively limited compared to full-shot training. However, for entity types with common semantics such as date, few-shot training can significantly boost the performance, which justifies the strong generalizability of the model.

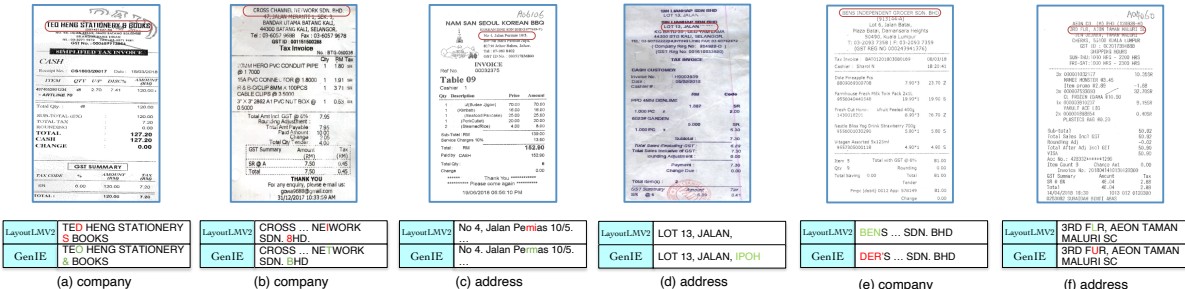

Figure 5: Qualitative examples of the entity extraction task on the SROIE dataset. We compare GenKIE with LayoutLMv2. In subfigures (a) to (d), GenKIE correctly fixes the OCR mistakes. In subfigures (e) to (f), GenKIE generates the wrong entity values even though there are no OCR errors. Best viewed in colour.

| Entity Type | 0 shot | 1 shot | 5 shot | 10 shot | full shot |
|---|---|---|---|---|---|
| company | 4.0 | 3.7 | 34.18 | 42.43 | 96.97 |
| address | 3.9 | 3.6 | 12.38 | 55.60 | 97.22 |
| date | 3.2 | 14.53 | 77.21 | 78.21 | 97.45 |
| total | 5.0 | 4.9 | 45.92 | 66.33 | 97.83 |

Table 6: Results of zero/few-shot entity extraction on SROIE.