# OpenReview forum: "GenKIE: Robust Generative Multimodal Document Key Information Extraction"
_EMNLP/2023/Conference — EMNLP 2023 Findings_

### Official Review · Reviewer_FmZH · 2023-07-31

**Soundness:** 2

**Excitement:**

3: Ambivalent: It has merits (e.g., it reports state-of-the-art results, the idea is nice), but there are key weaknesses (e.g., it describes incremental work), and it can significantly benefit from another round of revision. However, I won't object to accepting it if my co-reviewers champion it.

**Paper Topic And Main Contributions:**

This paper proposes to use a generative approach to perform key-value extraction of document images, e.g. decoding the key and value text directly. Compared to previous discriminative works that produces BIO tagging, this approach can fix OCR errors by design.

**Reasons To Accept:**

The proposed approach is more capable in theory.

**Reasons To Reject:**

While the generative approach is proposed to solve some OCR error issues, but the overall end-to-end accuracy is lower than previous methods.

**Reproducibility:**

3: Could reproduce the results with some difficulty. The settings of parameters are underspecified or subjectively determined; the training/evaluation data are not widely available.

**Reviewer Confidence:**

2: Willing to defend my evaluation, but it is fairly likely that I missed some details, didn't understand some central points, or can't be sure about the novelty of the work.

---

### Official Review · Reviewer_Y6Ra · 2023-08-06

**Soundness:** 4

**Excitement:**

4: Strong: This paper deepens the understanding of some phenomenon or lowers the barriers to an existing research direction.

**Paper Topic And Main Contributions:**

Authors propose a generative approach to key information detection in document images.
Instead of the standard OCR-followed-by-NER approach, they build a multimodal representation of the document by fusing image and text representations and then using generative approach to output the desired text. The generative approach helps in avoiding issues with
OCR errors. Results are presented on open datasets.

**Reasons To Accept:**


The multimodal approach to representing documents and using generative AI to avoid OCR errors is appealing.
There is very less work in this area and this paper helps move the field forward.

**Reasons To Reject:**

None.

**Reproducibility:**

4: Could mostly reproduce the results, but there may be some variation because of sample variance or minor variations in their interpretation of the protocol or method.

**Reviewer Confidence:**

5: Positive that my evaluation is correct. I read the paper very carefully and I am very familiar with related work.

---

### Official Review · Reviewer_r4Eh · 2023-08-09

**Soundness:** 3

**Excitement:**

4: Strong: This paper deepens the understanding of some phenomenon or lowers the barriers to an existing research direction.

**Paper Topic And Main Contributions:**

This paper describes a novel modeling approach called GenKIE for key information extraction tasks from scanned documents. Different from traditional sequence tagging approaches, the proposed approach adopts an encoder-decoder transformer to directly generate the desired data text. Experimental results on three datasets show that this new approach can achieve similarly good performance as prior work, and can be more robust to OCR noises in the input compared to other methods.


**Questions For The Authors:**

- Question A: (line 446) It is unclear what types of OCR noises are added in the ablation in Sec. 6.2. It would be good to clarify them in the appendix.
- Question B: (figure 4 and line 551) In the examples, do you show the original OCR text or perturbed version in Sec 6.2?
- Question C: (line 383) In the decoder, apart from the prefix decoding, do you use any constrained decoding methods (e.g., limiting the vocabulary to those appeared in the input)?


**Reasons To Accept:**

1. Overall this is a nice approach that can be helpful in practical tasks with potentially high OCR errors. In addition, the decoder-based training removes the need for token level annotations and reduces the cost for annotation.
2. While the idea of using an encoder-decoder network for KIE tasks is not completely new (as the author clarified in the paper (line 207)), it is unclear of the used prompts and the generation task in the previous work TILT. It is good that this paper provides a detailed explanation of the methodology and prompts and detailed analysis.


**Reasons To Reject:**

This work can be improved in the following dimensions:
1. While it is understandable, the size of the test set for all the datasets is generally small (e.g., FUNSD has only 50 examples) and the results might be highly volatile.
2. The design of the robustness experiment (line 458) can be improved. The argument from the authors is that the GenKIE approach can be robust to (realistic) OCR noises. However, the authors added additional OCR noise to the SROIE dataset (which already contains OCR noise). As such, this setting could be overly harsh to token-classification based models like LayoutLMv2: the experiment is based on the assumption that the current OCR models can have worse accuracy which is not necessarily true. An ideal experiment should be comparing two scenarios: (1) using the imperfect OCR text generated (from different OCR engines) vs. (2) using a manually corrected text representation in the input. By doing so it can reveal to what extent this approach can be helpful in practice.


**Reproducibility:**

4: Could mostly reproduce the results, but there may be some variation because of sample variance or minor variations in their interpretation of the protocol or method.

**Reviewer Confidence:**

4: Quite sure. I tried to check the important points carefully. It's unlikely, though conceivable, that I missed something that should affect my ratings.

**Typos Grammar Style And Presentation Improvements:**

- line 60: There is a missing reference for BROS.

---

### Official Review · Reviewer_Bc8K · 2023-08-12

**Soundness:** 3

**Excitement:**

3: Ambivalent: It has merits (e.g., it reports state-of-the-art results, the idea is nice), but there are key weaknesses (e.g., it describes incremental work), and it can significantly benefit from another round of revision. However, I won't object to accepting it if my co-reviewers champion it.

**Paper Topic And Main Contributions:**

The paper raises the issue of the standard token-classification-based approach to VDU being prone to OCR errors and proposes to use a generative sequence-to-sequence framework, which does not require costly token-level annotations and has the potential to correct OCR errors.

They augment an image-to-text model OFA with textual and layout features and finetune it on SROIE, CORD and FUNSD datasets, demonstrating competitive results against a panel of baselines.

In a follow up study they show their model's robustness to increasing OCR noise compared to LayoutLM and include ablations studies with different types of prompts.

**Questions For The Authors:**

Question A: It would be useful to have a reference of the fraction of labels affected by OCR corruption in the datasets.

Question B: Not clear how prompts can have visual features unless they are contained in the input document, which seems unlikely if the prompt starts with an entity type rather than value.

**Reasons To Accept:**

The motivation of standard discriminative extraction being prone to OCR noise is valid, and using a generative prediction framework is a sensible idea, if not particularly novel.

The presented generative system is competitive with several discriminative and few generative systems on 3 IE datasets. It especially excels in the robustness study where authors vary the level of OCR noise.

**Reasons To Reject:**

The idea of using encoder-decoder for IE is far from novel, which the authors partially acknowledge in the related work section.

They cite the limitations of a similar encoder-decoder Donut in related work but do not include it in the evaluation.

Also, not clear what innovations were performed on top of OFA other than adding textual and layout embeddings.

The authors go to a great length to explain the details that seem fairly common among multimodal transformers (e.g., concatenation of position embeddings with token embeddings, using ResNet feature patches), rather than focusing on specific modifications.

Vanilla OFA performance not included in the evaluation (or in the appendix), and it remains unclear whether or not the introduced textual features are effectively leveraged by the model.

Finetuning dataset sizes appear too small to expect significant improvement without dedicated textual pre-training.

**Reproducibility:**

4: Could mostly reproduce the results, but there may be some variation because of sample variance or minor variations in their interpretation of the protocol or method.

**Reviewer Confidence:**

3: Pretty sure, but there's a chance I missed something. Although I have a good feel for this area in general, I did not carefully check the paper's details, e.g., the math, experimental design, or novelty.

**Typos Grammar Style And Presentation Improvements:**

Language/grammar:

018 entice the generation

050 extracting interested entities

271 positional embedding element-wisely

Missing citations for BROS:
179 (?)

---

### Meta-Review · Area_Chair_vycr · 2023-09-15

**Recommendation:** 3

**Metareview:**

The authors consider a sequence to sequence approach for key information extraction, taking multi-model data for encoding and yielding the output structures directly, thereby avoiding OCR errors. This is a dedicated study for solving a practical problem. While there are existing sequence to sequence methods, the authors address several important issues in prompt design, generation tasking and OCR errors etc.

---

### Decision · Program_Chairs · 2023-10-07

**Decision:**

Accept-Findings

**Comment:**

The authors consider a sequence to sequence approach for key information extraction, taking multi-model data for encoding and yielding the output structures directly, thereby avoiding OCR errors. This is a dedicated study for solving a practical problem. While there are existing sequence to sequence methods, the authors address several important issues in prompt design, generation tasking and OCR errors etc.